# Rewiring of Cancer Cell Metabolism by Mitochondrial VDAC1 Depletion Results in Time-Dependent Tumor Reprogramming: Glioblastoma as a Proof of Concept

**DOI:** 10.3390/cells8111330

**Published:** 2019-10-28

**Authors:** Tasleem Arif, Oriel Stern, Srinivas Pittala, Vered Chalifa-Caspi, Varda Shoshan-Barmatz

**Affiliations:** 1Department of Life Sciences, Ben-Gurion University of the Negev, Beer-Sheva 84105, Israel; tashu100@gmail.com (T.A.); orielste@post.bgu.ac.il (O.S.); pittala@post.bgu.ac.il (S.P.); 2National Institute for Biotechnology in the Negev, Ben-Gurion University of the Negev, Beer-Sheva 84105, Israel; veredcc@bgu.ac.il

**Keywords:** cancer stem cells, differentiation, glioblastoma, mitochondria, si-RNA, TSPO, VDAC1

## Abstract

Reprograming of the metabolism of cancer cells is an event recognized as a hallmark of the disease. The mitochondrial gatekeeper, voltage-dependent anion channel 1 (VDAC1), mediates transport of metabolites and ions in and out of mitochondria, and is involved in mitochondria-mediated apoptosis. Here, we compared the effects of reducing hVDAC1 expression in a glioblastoma xenograft using human-specific si-RNA (si-hVDAC1) for a short (19 days) and a long term (40 days). Tumors underwent reprograming, reflected in rewired metabolism, eradication of cancer stem cells (CSCs) and differentiation. Short- and long-term treatments of the tumors with si-hVDAC1 similarly reduced the expression of metabolism-related enzymes, and translocator protein (TSPO) and CSCs markers. In contrast, differentiation into cells expressing astrocyte or neuronal markers was noted only after a long period during which the tumor cells were hVDAC1-depleted. This suggests that tumor cell differentiation is a prolonged process that precedes metabolic reprograming and the “disappearance” of CSCs. Tumor proteomics analysis revealing global changes in the expression levels of proteins associated with signaling, synthesis and degradation of proteins, DNA structure and replication and epigenetic changes, all of which were highly altered after a long period of si-hVDAC1 tumor treatment. The depletion of hVDAC1 greatly reduced the levels of the multifunctional translocator protein TSPO, which is overexpressed in both the mitochondria and the nucleus of the tumor. The results thus show that VDAC1 depletion-mediated cancer cell metabolic reprograming involves a chain of events occurring in a sequential manner leading to a reversal of the unique properties of the tumor, indicative of the interplay between metabolism and oncogenic signaling networks.

## 1. Introduction

Glioblastoma multiforme (GBM), affecting glial cells, is the most malignant brain cancer in adults, with inevitable relapse [1,2,3]. Such tumors, associated with high morbidity and mortality, are resistant to conventional treatments [4]. GBM is characterized by extensive tumoral heterogeneity, resulting in a large range of variabilities, including the transiently quiescent self-renewing cancer stem cells (CSC) [5]. The CSC hypothesis, whereby a hierarchical organization exists in a tumor, with phenotypically different sub-populations of tumorigenic CSCs, characterized by pluripotentic and multipotentic stem cell markers and their non-tumorigenic descendants, can explain the concept of tumor heterogeneity [6,7,8]. CSCs seem to have extraordinary resistance capabilities to conventional treatments, such as chemotherapeutics and radiotherapy [9]. Cancer cell proliferation often relies on aerobic glycolysis, (Warburg effect) [10], reflected in a high rate of glucose uptake, lactate production, and acidification of the tumor microenvironment [11]. However, in addition to glycolysis, GBM cells also use the tricarboxylic acid cycle and oxidative phosphorylation as sources of energy [12].

The metabolic reprograming of cancer cells is regulated by genes and factors that modulate the cellular response to changes in the tumor microenvironment [13]. Indeed, mutations in metabolic enzymes or changes in signaling pathways regulating metabolism have been identified [14]. As such, growing insight into tumor metabolic flexibility may lead to new potential treatments for GBM [15].

The mitochondrial protein VDAC1, a multifunctional large channel (pore diameter of 3 nm), plays an important role in energy production, serving as the “gate” for metabolic and survival signals generated elsewhere in the cell. VDAC1, located at outer mitochondrial membrane (OMM), allows transport of nucleotides and metabolites (up to 5 kDa), fatty acids, and cholesterol and ions, including Ca^2+^, across the OMM [16,17,18]. As such, VDAC1 is highly expressed in various tumors obtained from patients or produced in mouse models [16,17,18], pointing to its pivotal role in regulating cancer cellular energy and metabolism. The significance of VDAC1 for cancer cell viability is underscored by results showing that abrogation of VDAC1 expression reduced cellular ATP levels, tumor development and growth [16,18,19]. In fact, we have demonstrated that VDAC1 is a novel target for cancer therapy [16,17,18,20].

In our previous study [16], using several GBM cell lines (U-87MG, U-251MG, U-118MG, LN-18,C6, and GL-261) and human GBM patient-derived cells as MZ-18 and MZ-327 and the glioma-derived stem cell line G7, we demonstrated that silencing the expression of VDAC1 using specific si-RNA (si-hVDAC1) dramatically inhibited cell growth. Using a subcutaneous (s.c.) or intracranial-orthotopic GBM model, we showed that that silencing the expression of human VDAC1 using si-hVDAC1, dramatically decreased tumor growth [16]. Moreover, the remaining small “tumor” exhibited inverted oncogenic properties, such as rewired metabolism, as well as inhibited angiogenesis, epithelial–mesenchymal transition (EMT), stemness and invasiveness, in addition to differentiation into cells expressing protein markers of neurons and astrocytes [16]. These effects mediated by VDAC1 depletion are associated with alterations in signaling pathways that allow for attack on the interchange between cell metabolism and cancer signaling systems leading to the reprograming of tumor cells, including their microenvironment [16]. Similar results were obtained for mouse models for lung cancer and triple negative breast cancer [16,18,19].

Recently, we showed in cultured cells of GBM, lung cancer and triple negative breast cancer that although VDAC1 silencing occurred within a day, the cells underwent reprograming with respect to rewiring metabolism, elimination of CSCs, alteration of transcription factor (TF) expression and differentiation, were developed with time and maximal changes were observed after 3 weeks of silencing VDAC1 expression [21].

In the present study, using immunohistochemistry, immunoblotting, quantitative real-time PCR (qRT-PCR) and proteomics profiling by mass spectroscopy, we demonstrate that tumor reprograming is dependent on the duration that the tumor was depleted of VDAC1. The global changes in the expression of proteins associated with signaling, protein synthesis and degradation, and DNA structure and replication, epigenetic and differentiation were pronounced after long-term treatment with si-hVDAC1 and follow the metabolic reprograming.

## 2. Materials and Methods

See the Appendix A for detailed Materials and Methods.

### 2.1. Xenograft Experiments

The si-RNA specific to human (h)VDAC1 (si-hVDAC1) and non-targeting (si-NT) were obtained from Genepharma. The following sequences were used, with 2′-O-methyl-modified nucleotides indicated in bold and underlined (nucleotide positions are provided for sense (S) and anti-sense (AS) sequences): si-hVDAC1/2A: S: 238-5′-ACAC**U**AG**G**CACC**G**AGA**U**UA-3′-256 and AS: 238-5′-UAAUC**U**CGGUGCCUA**G**UGU-3′. si-NT: S: -5’-GCAAACAUCCCAGAGGUAU-3’ and AS: 5’-AUACCUCUGGGAUGUUUGC-3’.

U-87MG (2 × 10^6^) cells were inoculated s.c. into the hind leg flanks of athymic eight-week-old male nude mice (Envigo). Fourteen days post-inoculation, when tumor volumes had reached 50–80 mm^3^, the mice were randomized into two groups (8 animals/group), treated with non-targeting si-RNA (si-NT) or si-hVDAC1 mixed with in vivo transfection reagent JetPEI and injected into the established s.c. tumors (50 nM final concentration, 2 boluses) every three days. One group received treatment for 19 days (short-term treatment) and the second group received treatment for 40 days (long-term treatment). At the end of the experiments, the mice were sacrificed, tumors were excised, and half of each tumor was either fixed and processed for immunohistochemistry (IHC) or frozen in liquid nitrogen for later immunoblot and RNA isolation. Experimental protocols were approved by the Institutional Animal Care and Use Committee.

### 2.2. Immunoblotting and Immunohistochemistry (IHC) and Immunofluorescence (IF) Staining

Tumor protein extracts, SDS-PAGE and immunoblotting, IHC, and IF methods were performed as described in the Appendix A section, using the antibodies listed in Appendix A, along with their sources and the dilutions used.

### 2.3. RNA Preparation, qRT-PCR, Liquid Chromatography–High–Resolution Mass Spectrometry (LC-HR-MS/MS) and Proteomics Analysis

Total RNA was isolated from liquid nitrogen-frozen tumors obtained from mice treated with si-NT or si-hVDAC1 for short (19 days) or long )40 days) periods, as described in the Appendix A section. The isolated RNA was used for qRT-PCR, performed as described in the Appendix A section using specific primers listed in Appendix A.

Mass spectrometry (MS)-based proteomics profiling and initial processing of the results were carried out at the de Botton Institute for Protein Profiling, G-INCPM, Weizmann Institute of Science. For LC-HR MS/MS analysis, proteins were extracted from liquid nitrogen-frozen tumor tissues obtained from mice treated with si-NT or si-hVDAC1 for short or long periods (see Appendix A section) and stored at -80 0C until LC-HR-MS/MS analysis. Samples were subjected to tryptic digestion, alkylation, detergent removal and desalting and then subjected to LC-HR MS/MS analysis, described in the Appendix A section.

### 2.4. Statistical Analyses for Identification of Differentially Expressed Proteins

Human proteins for which at least one unique peptide was identified (n = 1845) were subjected to further analysis. The bioinformatics analysis was performed at the National Institute for Biotechnology in the Negev (NIBN) Bioinformatics Core Facility using Partek Genomics Suite, Perseus, and Excel for additional processing. Quality assurance of the results using principal component analysis (PCA), signal distribution plots and hierarchical clustering showed that sample 1 of the 3 samples subjected to long-term treatment with si-NT or si-hVDAC1 were exceptional and was, therefore, excluded from further analysis, leaving duplicate long-term samples and triplicate short-term-treated tumors samples. Prior to statistical testing, intensities were transformed to log2 scale.

The existence of many zero intensities in the dataset poses a challenge for the ANOVA statistical tests. Two approaches were used to circumvent this challenge: (1) zero intensities were replaced by “missing values” and thus neglected (dataset 1) and (2) zero intensities were imputed (replaced) by random numbers derived from a normal distribution in the low expression range using default Perseus parameters (width = 0.3, down shift = 1.8). Imputation was repeated five times to avoid relying too heavily on fabricated numbers (datasets 2–6).

For each of the six resulting datasets, one-way ANOVA was performed considering four groups: short-term treatment with si-NT or si-hVDAC1 and long-term treatment with si-NT or si-hVDAC1, and two contrasts: short-term treatment with si-hVDAC1 or si-NT and long-term treatment with si-hVDAC1 or si-NT. Proteins were considered to be differentially expressed if they had a nominal *p*-value < 0.05 and an absolute fold of change (FC) higher than 1.5 either in dataset 1 or in more than 3 of the imputed datasets.

Hierarchical clustering was performed in Partek, using Pearson’s dissimilarity and complete linkage. Enrichment analysis versus gene ontology (GO) biological process and GO cellular component assignment was performed in Enrichr [22] using Fisher’s exact test.

### 2.5. Statistics

Results are presented as the means ± SEM of results obtained from independent experiments are presented. A difference was considered statistically significant when the *p*-value was deemed < 0.05 (*), < 0.01 (**) or < 0.001 (***), assessed through an unpaired Student’s two-tailed t-test.

## 3. Results

Modified metabolism is one of the hallmarks of cancer [23]. VDAC1, a protein that regulates cell metabolism, is highly expressed in various cancers, including GBM [16], pointing to its important function in cancer development and survival. Previously, we demonstrated that VDAC1 depletion in GBM using si-RNA resulted in inhibited tumor growth involving rewired metabolic properties and inhibited cell proliferation, EMT, invasion, angiogenesis, and stemness, leading to the appearance of neuronal-like cells [16,19].

Here, we asked whether such tumor reprograming is dependent on the duration of treatment with si-RNA specific to human VDAC1 (si-hVDAC1). Accordingly, we treated a mouse model containing a human GBM xenograft derived from U-87MG cells with si-hVDAC1 every three days for 19 days (short period) or for 40 days (long period) and followed reprograming of the treated residual “tumor” with respect to metabolism, CSCs, and differentiation using immunoblotting, qRT-PCR and proteomics profiling (mass spectroscopy). 

### 3.1. VDAC1 Depletion Inhibits Cancer Tumor Development and Induced Metabolic Reprograming Following Both Short- and Long-Term Treatment

The experimental protocol for testing the effect of short- and long-term treatment with non-targeting si-RNA (si-NT) or si-hVDAC1 is presented in Figure 1A. Silencing VDAC1 expression in U-87MG cell-derived tumors by si-hVDAC1 led to marked decreases in tumor growth following short- or long-term treatment with the si-RNA (Figure 1B,C). Injecting mice every three days with si-NT or si-hVDAC1 revealed that in si-NT-treated tumors (TTs), tumor volume grew exponentially, increasing 14- and ~45-fold after short- and long-term treatment, respectively. In contrast, growth was markedly inhibited in si-hVDAC1-TTs, which only increased in size 3- and 8-fold after short- and long-term treatments, respectively (Figure 1B,C). All mice were sacrificed after the end of treatment, tumors were excised and fixed, and sections were immunostained for metabolism-related proteins, CSCs, differentiation, and TSPO.

Alterations in metabolism that occurred during cancer development involve a spectrum of functional aberrations and mutations which contribute to enhanced glycolysis and elevated expression levels of glucose transporters (Glut) and glycolytic enzymes [24]. Immunoblotting with protein specific antibodies showed that as expected, si-hVDAC1-TTs showed dramatic decreases in VDAC1 expression levels after both short- and long-term treatment with si-hVDAC1 (Figure 1D–G). si-hVDAC1-TTs also showed dramatic decreases in the expression levels of Glut-1, hexokinase (HK-I), glyceraldehyde dehydrogenase (GAPDH) and lactate dehydrogenase (LDH), as compared to their levels in si-NT-TTs. Expression levels of the Kreb’s cycle enzyme citrate synthase (CS) and the mitochondrial electron transport complex IVc (Complex IVc) were also highly reduced in si-hVDAC1-TTs, in agreement with changes in the expression of enzymes of the oxidative phosphorylation (OXPHOS) (Figure 1D–G). Similar results, including decreases in the level of ATP synthase subunit 5a (ATP syn5a), were obtained by qRT-PCR (Figure 1H,I).

IHC analysis of selected metabolically related proteins, such as Glut1, VDAC1, and CS, demonstrated a similar result, with a 7-14-fold decrease in the levels of expression (Figure 1J–M). The decreased expression of Kreb’s cycle and OXPHOS enzymes agrees with the concept that cancer cells mainly use a combination of glycolysis and mitochondria to produce energy, reflecting the prevailing normoxic or hypoxic conditions in a tumor [25].

In general, both short- and long-term treatment with si-hVDAC1 similarly reduced the expression of metabolism-related enzymes, pointing to VDAC1 depletion in cancer cells reversing their reprogramed metabolism.

### 3.2. Tumor Treatment with si-hVDAC1 Decreases CSCs and Increases Expression of Differentiation-Associated Proteins

Glioma CSCs as neural stem cells (NSCs), express stem cell markers such as CD133, SOX2, KLF4, and Nestin. The expression of GBM stem cells markers after short- and long-term treatment of si-hVDAC1-TTs was analyzed by immunoblotting and qRT-PCR (Figure 2). si-hVDAC1 tumor treatment markedly decreased the expression of CSCs markers, such as CD133, SOX2, KLF4, Nestin, and CD44, as evaluated by quantitative immunoblotting (Figure 2A–D) and qRT-PCR (Figure 2E,F). qRT-PCR also revealed a major decrease (3–40-fold) in mRNA levels of the transcription factors OCT3/4, SOX2, and Nanog following both short- and long-term treatment, suggesting a decrease in CSCs levels upon metabolic reprograming.

The apparent clearance of CSCs in tumors upon treatment with si-hVDAC1 may result from arrested cell proliferation and/or promoted differentiation. We previously demonstrated that si-hVDAC1-treated GBM tumors underwent differentiation [16,19]. Here, we evaluated whether this differentiation is dependent on the duration of si-hVDAC1 treatment (Figure 3). Accordingly, we examined the expression levels of several differentiation indicators associated with the nervous system in si-NT-TTs and si-hVDAC1-TTs subjected to short- or long-term treatment (Figure 3). Immunostaining for the mature astrocyte marker glia fibrillary acidic protein (GFAP) and neuron markers, such as microtubule-associated protein 2 (MAP-2), tubulin beta-III (TUBB3) and glutamate decarboxylase 1 (GAD1/GAD67), involved in gamma-aminobutyric acid (GABA synthesis, revealed similarly low expression levels in si-NT-TTs and si-hVDAC1-TTs subjected to the short-term treatment (Figure 3A,C). However, their expression was highly increased in tumors subjected to long-term si-hVDAC1 treatment (Figure 3B,C). This was further reflected in the qRT-PCR data, showing 15- to 40-fold increases in si-hVDAC1-TTs subjected to long treatment (Figure 3D). Similar results were obtained by immunofluorescence (IF) staining for astrocyte, using anti-GFAP and for neurons, using anti-TUBB3 antibodies, showing high increases in their expression levels in the long-term si-hVDAC1-TTs (Figure 3E–G).

These results indicate that differentiation of cancer cells/CSCs into neuronal-like cells is a process requiring longer times during which the cells are depleted of VDAC1. The findings are, moreover, in agreement with the proposal that altering cancer metabolism affects cell differentiation [26].

### 3.3. VDAC1 Depletion in GBM Xenografts Does Not Induce Cell Death but Results in an Increase in the Expression of Pro-Apoptotic Proteins, and a Decrease in TSPO Expression

As shown previously [16,19], VDAC1 depletion did not induce cell death in the si-hVDAC1-TTlines as reflected in the terminal deoxynucleotidyl transferase dUTP nick end labeling (TUNEL) staining (Figure 4A). Following our previous study [1] that revealed an increase in the levels of several pro-apoptotic proteins after VDAC1 depletion, we were interested to discover whether this increase was affected by the duration of treatment.

IHC evaluation of the expression levels of cytochrome *c* (Cyto *c*), caspase 3, and apoptosis-inducing factor (AIF) indicated a dramatic increase in the levels of Cyto *c* and caspase 3, in both short (11- and 7-fold) and long-term si-hVDAC1-TTs (17- and 9-fold), respectively (Figure 4B,C,E). This increase in the expression of pro-apoptotic proteins may be related to their alternative, non-apoptotic functions, ranging from bioenergetics to differentiation, metabolism, and inflammation [27,28,29,30,31]. Interestingly, Cyto *c* and caspase 3 were also observed in the nucleus (Figure 4B,C, inset).

In contrast, AIF expression levels decreased 10- and 15-fold in short- and long-term si-hVDAC1-TTs, respectively (Figure 4D,E). Although, as an inducer of cell death, AIF possesses anti-neoplastic potential, the molecule is over expressed in cancers [1] and elevated AIF protein levels may benefit tumorigenesis, suggesting that AIF may fulfill a pro-survival function [1,32].

AIF is expected to translocate to the nucleus and trigger DNA degradation only after the induction of apoptosis when the molecule is released from mitochondria [33]. The results presented here in Figure 4D, inset revealed that AIF is localized to the nucleus with no apoptosis. This may suggest that the non-apoptotic tumor pro-survival activity of AIF requires a nuclear-based activity.

### 3.4. TSPO Is Localized to Both the Mitochondria and the Nucleus and VDAC1 Depletion in GBM Xenografts Decreases the Expression of TSPO

TSPO is an OMM protein mediating multiple functions, including cholesterol import, steroid synthesis, regulation of mitochondrial metabolism, and apoptosis [34]. Several studies have addressed the expression levels of TSPO in tumors [35,36,37,38]. Analysis of TSPO ligand binding and mRNA levels revealed a high level of TSPO expressed in various types of cancer [35,36,37,38].

Here, using IHC of healthy and cancer tissue arrays and quantitative analysis, we demonstrate that the levels of TSPO were 10-fold higher in brains from patients with GBM than in healthy tissue (Figure 5A). In addition to GBM, other tumor types including stomach, lung, hepato-carcinoma (liver), kidney, and breast, also overexpress TSPO (10- to 20-fold) relative to the levels in healthy tissue, (Figure 5B). Similar results were obtained with chronic lymphocytic leukemia (CLL) (Figure 5C,D). These observations suggest that TSPO is involved in the modulation of cell proliferation and tumorigenesis.

Next, we analyzed by IHC the expression of TSPO in U-87MG xenografts, showing that the expression of TSPO decreased by about 80% in si-hVDAC1-TTs, whether they were treated for a short or long period (Figure 6A,B). These findings suggest that tumor VDAC1 depletion resulted in reduced TSPO levels and thereby decreasing the TSPO tumor pro-survival activity.

These results were confirmed by immunofluorescence staining with anti-TSPO antibody, which revealed a decrease in the expression of TSPO after VDAC1 depletion in both short- and long-term TTs (Figure 6C,D). In addition, the co-immunostaining with anti-VDAC1 antibodies (Figure 6C,D) indicated that the proteins co-localized as expected for proteins in the same membrane, the OMM. Interestingly, in the si-NT-TTs, TSPO also co-localized with DAPI staining of the nucleus, suggesting that in tumors that overexpress TSPO, the protein translocates to the nucleus (Figure 6C,D, enlargements).

As TSPO is overexpressed in GBM tissues from patients (Figure 5), the nuclear location of TSPO is expected, (Figure 7A). Similarly, the presence of TSPO in the nuclei of U-87MG xenografts treated with si-NT was demonstrated by IF and IHC staining (Figure 7B,C).

As the nuclear localization was addressed in several studies and TSPO was not found to be localized in the nucleus of GBM U-118 cells in culture [39], we analyzed whether TSPO is localized in U-87MG cells in culture. Our results demonstrate no significant localization of TSPO in the nucleus of U-87MG cells (Figure 7D). The nuclear localization of TSPO in the U-87MG xenografts (Figure 7C) but not in cells in culture (Figure 7D), therefore suggests that the conditions in the tumor, either in the mouse model or in GBM patients (Figure 7A) are responsible for inducing TSPO translocation to the nucleus.

### 3.5. Mass Spectrometry Analysis of the GBM Tumor Protein Profile after Short- and Long-Term si-hVDAC1 Treatment

To identify proteins showing different expression levels in GBM tumors from the four groups, namely si-NT- and si-hVDAC1-treated for short or long periods, tumors were subjected to LC-HR MS/MS, proteomics, and functional enrichment analyses. After filtering for human proteins which had at least one unique peptide, about 1845 proteins were submitted for subsequent analysis. The differentially expressed proteins (*p*-value < 0.05 and fold change |FC| > 1.5) between si-NT-TTs and si-hVDAC1-TTs numbered 25 and 146 in the short- and long-term-treated groups, respectively, with 60% and 86% of the proteins being down-regulated, respectively (Figure 8A). 

The hierarchical clustering of the differently expressed proteins in short- and long-term TTs (Figure 8B) and the volcano plots (Figure 8C,D) showed that larger numbers of the differentially expressed proteins were down-regulated, with their numbers increasing in the long-term-treated group. Some of the proteins whose expression was highly up- or down-regulated with the highest *p*-value are indicated and are further discussed below.

Next, enrichment analysis of the proteins differentially expressed between the si-NT-TTs and si-hVDAC1-TTs after long-term treatment was performed using the GO databases for biological processes (Figure 8E) and cellular components (Figure 8F) [40,41]. Such analysis revealed alterations in the expression levels of proteins assigned roles in metabolism, such as in the respiratory electron chain (ETC) and mitochondrial ATP synthesis, cellular macromolecular synthesis, and DNA metabolic processes, as well as in the expression of proteins associated with intracellular trafficking (Golgi vesicle and ER-Golgi transport), replication, interferon–γ-mediated signaling, cellular response to DNA damage and mRNA transport (Figure 5E). The differentially expressed proteins were mainly associated with mitochondria (inner and outer membranes and matrix), components of the nuclear chromosome and telomeric regions, focal adhesions, the nucleus and the endoplasmic reticulum (ER) lumen (Figure 8F). These results were further grouped according to their function and are discussed below.

### 3.6. Differential Expression of Metabolism-, Transport-, and Trafficking-Related Proteins after Short- and Long-Term si-hVDAC1 Treatment of GBM Tumors

Proteomics analysis showed that si-hVDAC1-TTs metabolic reprograming was also reflected in the modified expression of several metabolism-related proteins (Figure 9A, Appendix A). These included ALDH2 (aldehyde dehydrogenase) and TIGAR (fructose-2,6-bisphosphatase), a negative regulator of glycolysis [42], the levels of which unchanged or increased by 2-fold following the short si-hVDAC1 treatment, yet were further increased 9- and 38-fold, respectively, following longer treatment. On the other hand, the expression of GPDIL (glycerol-3-phosphate dehydrogenase 1-like protein), catalyzing glycerol 3-phosphate production, ABDH10 (mycophenolic acid acyl-glucuronide esterase) and NDUFS8 (NADH dehydrogenase), a subunit of the complex I respiratory chain, decreased 2–4-fold and 10–42-fold after short- and long-term treatment, respectively. The proteomics data explored additional proteins related to metabolism, with some, such as TIGAR, an inhibitor of glycolysis, being increased. Other proteins related to energy production, such as subunits of the electron transport chain, were greatly decreased. These results suggest that metabolism reprogramming develops with the time of tumor VDAC1 depletion.

Another protein group related to metabolism are those mediating the transport of metabolites (Figure 9B, Appendix A), such as Glut1 (glucose transporter) and ARL6IP1 (ADP-ribosylation factor-like protein 6-interacting protein 1). ARL6IP1, a regulator of the excitatory amino acid carrier 1 (EAAC1) via increasing its affinity for glutamate, which its expression levels were unchanged or decreased 2-fold in tumors treated for a short period with si-hVDAC1 and further decreased 10- and 20-fold after long-term si-VDAC1-treatment. In addition, while the expression of the zinc transporter SLC39A14 (zinc transporter ZIP14) decreased about 5-fold, the expression level of the mitochondrial Ca^2+^/H^+^ antiporter protein (LETM1) [43] increased 4-fold in the long-term-treated si-hVDAC1-TTs.

The expression levels of several intracellular trafficking-related proteins were highly changed in the long-period si-hVDAC1-TTs, as compared to their levels in the short-term si-hVDAC1-TTs (Figure 9B, Appendix A). The expression levels of the Golgi and endosome protein ARFGEF2 (Brefeldin A-inhibited guanine nucleotide-exchange protein 2, BIG2), a Rho GTPase regulatory protein involved in dendrite growth and maintenance [44], were reduced close to 50-fold in the long-term TTs (Figure 9B, Appendix A). ARFGEF2 specifically activates class I ARFs (ARF1 and ARF3) in vivo to activate trafficking of vesicles from the Golgi apparatus and endosomal compartments [45]. The expression of EXOC1 (exocyst complex component 1)*,* which plays an important role in the tethering step of exocytosis [46], was also decreased about 25-fold in the long-term si-VDAC1-TTs. Expression of the nuclear pore complex protein NUP88 was also reduced (~19-fold) in the long-term si-VDAC1-TTs. Over-expression of NUP88 was observed in many tumors and has been experimentally proven to promote tumorigenesis [47,48]. The expression levels of GOLGA5 (Golgin subfamily A member 5), involved in maintaining the Golgi and in intra-Golgi retrograde transport, decreased 2-fold in the short-term TTs and 16-fold in the long-term si-VDAC1-TTs. Similarly, the expression levels of NUP93 (a nuclear pore complex protein), playing a role in the nuclear pore complex assembly and/or maintenance, and of PSD3 (PH and SEC7 domain-containing protein 3), a guanine nucleotide-exchange factor for ARF6, were unchanged in the short-term si-VDAC1-TTs but decreased 10-fold in the long-term si-hVDAC1-TTs. The expression of RAB12 (a member of the RAS oncogene family), which plays an important role in vesicle transport and trafficking within cells [49] and regulates the degradation of transmembrane proteins (as transferrin receptor 1) in different cellular compartments (Golgi complex, endosomes and lysosome), decreased about 2- and 8-fold in the short-term and long-term si-hVDAC1-TTs, respectively (Figure 9B, Appendix A). si-hVDAC1 treatment for a short term had no significant effect on GOPC (Golgi-associated PDZ and coiled-coil motif-containing protein) expression levels, although long-term treatment increased its level about 4-fold (Figure 9B, Appendix A). GOPC inhibits MAPK (a mitogen-activated protein kinase) activation, interacts with c-ros-oncogene 1 (ROS) and controls intracellular protein trafficking [50].

These results indicate that the expression of transport- and trafficking-associated proteins was strongly inhibited in tumors subjected to a long period of VDAC1 depletion, relative to short-term si-hVDAC1-TTs. This points to reduced transport and trafficking activity following a long period of tumor VDAC1 depletion.

### 3.7. Differential Expression of Signaling-, Development-, Differentiation-, and Human and Mouse Microenvironment-Related Proteins in GBM Tumors after Short- and Long-Term si-hVDAC1 Treatment

Tumorigenesis is the result of the complex regulation and interplay of different signal transduction pathways. As si-hVDAC1 treatment reduced tumor growth, reprogramed cancer cell metabolism, and induced differentiation, we expected alterations in signaling pathways upon such treatment. Indeed, (Figure 10, Appendix A), the expression levels of RRAD (Ras-related associated diabetes), which functions in hepatocellular carcinoma as a metastasis suppressor [51], and of STAT6 (signal transducer and activator of transcription 6), which activates gene transcription in response to cytokines [52], increased 2-3 fold after a short-term treatment with si-hVDAC1 but over 12-fold following long-term treatment with si-hVDAC1 (Figure 6, Appendix A). The expression level of SUSD2 (SUShi domain-containing 2), a Notch3-regulating protein that participates in the orchestration of cell adhesion and migration [53], increased over 100-fold (Figure 10, Appendix A).

On the other hand, the expression of other signaling molecules was down-regulated following si-hVDAC1 treatment (Figure 10A, Appendix A). These included ICAM1 (CD54) (intercellular adhesion molecule 1), which mediates adhesion of circulating leukocytes to the blood vessel wall and activated endothelium [54] and contributes to tumor metastasis [55]. PLXNB2 (plexin B2) that functions as an activator of GTPase Rho [56] tumor cell proliferation and invasion [57] and mediates angiogenesis-supporting tumor growth [58] and tumor metastasis [55], and LANCL2 (lanthionine synthetase C-like protein 2), a signaling pathway protein that is expressed in the plasma and nuclear membranes of immune cells and neurons, as well as in the gastrointestinal tract, testis and pancreas [59]. While short-term si-hVDAC1 treatment slightly attenuated their expression levels, long-term treatment further decreased their expression, with PLXNB2, LANCL2, and ICAM1 expression levels being reduced 5-, 10-, and 16-fold, respectively.

As we found that si-hVDAC1 GBM tumor treatment led to cell differentiation and the appearance of neuron-like cells (Figure 3), we assessed whether the differentiation of cancer cells into neuron-like cells could be reflected in the proteomic data and if so, whether it was treatment duration-dependent.

The expression of neuropilin 1 (NRP1), an adhesion molecule that serves as receptor for class 3 semaphorin A (SEMA3), and of vascular endothelial growth factor (VEGF), inhibits cell proliferation and induces apoptosis [60], were not significantly changed after short-term treatment with si-hVDAC1 but were reduced about 5-fold after long-term treatment (Figure 10A, Appendix A). Rho-associated coiled-coil-containing kinase 1 (ROCK1) is a serine protein kinase, expressed mostly in immune cells and regulated by small GTP-binding protein RhoA, to affect cell proliferation, adhesion, inflammation, and oxidative stress [61]. The expression level of ROCK1 increased some 4-fold in the short-term si-hVDAC1-TTs, while long-term si-VDAC1-TTs showed a 14-fold decrease in ROCK1 expression (Figure 10A, Appendix A). This switch in expression after short- and long-term si-hVDAC1 treatment may be due to the multiple functions of this protein, which could be affected by other changes that take place during the long period of VDAC1 depletion.

These results suggest that rewiring of cell metabolism and signaling pathways leads to altered expression of proteins associated with cell maturation/differentiation.

The use of si-RNA specific to human VDAC1 allows for evaluating crosstalk between cancer cells and the tumor microenvironment. Cancer cells can recruit numerous cell types, such as tumor-associated macrophages (TAMs), to induce neo-vascularization, and extracellular matrix (ECM), which provides a structural scaffold. The ECM contains cellular proteases, such as cathepsins, matrix metalloproteinases, fibrous proteins, such as elastin, collagens, fibronectin and laminins, and proteins that function in cell-ECM interactions, such as heparanase, integrins and laminin [62].

As the microenvironment mainly consists of host components, we analyzed changes in the expression of mouse proteins in the short- and long-term si-hVDAC1-TTs (Figure 10B, Appendix A). The results show that tumor treatment with si-hVDAC1 led to microenvironment rearrangement, with changes in the expression of proteins associated with stromal activity (Figure 10B, Appendix A). The expression levels of myosin-10 (MYH10, Myo10, NM-IIB), and which plays a role in cell-matrix adhesion, cell-cell interaction, cell polarity, cell migration, and apoptosis of embryonic stem cell [63], increased 4- and 80-fold following short- and long-term treatment with si-hVDAC1, respectively. On the other hand, the expression levels of several proteins decreased. A 2-3-fold decrease in the expression of the cell surface glycoprotein CD47 was obtained upon short- and long-term treatment with si-hVDAC1. CD47 is expressed on all cell types and is involved in regulation of the survival and death of the cell and its differentiation [64] via it acting a as ligand for macrophage-expressed signal-regulatory protein alpha (SIRPα) [65]. However, CD47 is a widely expressed counter-receptor for the inhibitory phagocyte receptor SIRP in the tumor microenvironment, which limits cooperation between anti-tumor T cell immunity and radiation therapy, as its blocking enhances tumor radio-sensitivity [66]. Thus, the decrease in CD47 levels upon VDAC1 depletion would be expected to enhance anti-tumor immunity.

Embigin (EMB), a member of the immunoglobulin superfamily, mediates interactions between the cell and the extracellular matrix [67] and is connected to neuromuscular junction formation and plasticity [68]. EMB expression levels were unchanged after short-term treatment but decreased over 6-fold following long-term treatment with si-hVDAC1. The expression levels of HMGB2 (high mobility group protein B2) that promotes pro-inflammatory conditions [69], which is associated with poor prognosis of pancreatic cancer and which promotes cell proliferation and survival [70], were unchanged or decreased 7-fold following short-term and long-term treatment with si-hVDAC1, respectively. Similarly, proteins associated with ECM organization and modification, such as the matrix metalloproteinases (MMPs) MMP8 and MMP9, both zinc-dependent enzymes which degrade components of the extracellular matrix and basement membrane [71], showed 1.5-2-fold reduced expression following short-term treatment and 12- and 18-fold decreases after the long-term treatment with si-hVDAC1, respectively (Figure 10B, Appendix A).

We also analyzed proteins of human origin that are related to the tumor microenvironment (Figure 10C, Appendix A). The expression level of CD44, a receptor for hyaluronic acid (HA), matrix metalloproteases and collagens that can mediate cell-cell and cell-matrix interactions and cell adhesion [72], was further decreased in long-term TTs. The levels of glypican-1, a proteoglycan that influences cell proliferation, differentiation and gene expression [73] decreased about 4-fold in the long-term si-hVDAC1-TTs. The expression of proteins, such as ITGA3 and 5, were unchanged in the si-hVDAC1-TTs but decreased some 3-fold following treatment for long-term with si-hVDAC1 (Figure 10C, Appendix A).

These results show that the metabolic changes in cancer cells of human origin resulting from hVDAC1 depletion led to global changes, with impact on the expression of proteins associated with the microenvironment in the mouse cells.

### 3.8. Modified Expression of Proteins Associated with Protein Synthesis and Degradation, DNA Structure and Replication, and Epigenetic Regulation upon si-hVDAC1 Treatment

In tumors, proteins involved in protein synthesis and post-transcriptional modifications, DNA structure and replication, and epigenetic regulation associated with tumor growth and cancer cell proliferation are modified. Here, we identified several proteins related to these functions that showed modified expression levels upon si-hVDAC1 treatment, with such changes being highly enhanced following long-term si-hVDAC1 treatment.

Expression levels of several proteins associated with protein synthesis and degradation were modified (Figure 11A, Appendix A). Some proteins were unchanged in tumors treated for a short term with si-hVDAC1 but greatly decreased (6- to over 50-fold) following long-term treatment with si-hVDAC1. These proteins included the E3 ubiquitin-protein ligase midline-1 (MID1) that possesses E3 ubiquitin ligase activity towards different proteins, such as phosphatase PP2A [74], calpain [75] and NEDD8 ultimate buster 1 (NUB1), recruiting proteins for proteasome-mediated degradation [76]. In the case of the alpha subunit of polypeptide-associated complex subunit alpha (αNAC), a component of a ribosome associated complex that interacts with the nascent peptide to protect it from proteolysis and facilitates its folding, expression decreased 8-fold (Figure 11A, Appendix A). 

The expression levels of mitochondrial ribosomal proteins L18 (MRPL18), L37 (MLRP37) and L9 (MRPL9), all of which induce translation of mRNAs following translocation to mitochondria, and which include family members associated with cancer [77,78,79,80], were unchanged following short-term treatment, yet decreased following long-term shVDAC1 treatment (3-, 4-, and 25-fold, respectively) (Figure 11A, Appendix A).

Epigenetic regulation is crucial for tumor growth, metastasis and chemotherapy resistance [81]. The expression levels of several proteins associated with DNA structure and epigenetic regulation were modified upon si-hVDAC1 treatment (Figure 11B, Appendix A). Similarly, the expression levels of the human histone PARylation factor 1 (HPF1), facilitating histone serine ADP-ribosylation in response to DNA damage [82], as well as that of the heterochromatin protein chromobox homolog 5 (CBX5/HP1α), which interacts with heterochromatic and euchromatic promoters to suppress gene expression, were not significantly changed following short-term treatment with si-hVDAC, yet were reduced 15- and 9-fold, respectively, following longer treatment (Figure 11B, Appendix A).

Among DNA structure-related proteins, si-hVDAC1 treatment for a short term had a negligible impact on the expression of histone H1x (H1Fx), needed for structural maintenance of chromosomes protein 2 (SMC2), and non-SMC condensin I complex subunit H (NCAPH/CAP-H), both subunits of the condensin complex involved in regulating DNA damage. After long-term treatment, their expression levels were reduced 3.5-, 9-, and 25-fold, respectively (Figure 11B, Appendix A). In contrast, long-term treatment with si-hVDAC1 increased the expression level of H1F0 3-fold. 

DNA replication is a crucial regulator of cancer [83]. si-hVDAC1 treatment also altered the expression of DNA replication-related proteins (Figure 11C, Appendix A). The expression levels of the ribonucleoside-diphosphate reductase subunit M2 B (RRM2B), encoding p53-inducible small RNR (ribonucleotide reductase) subunit (p53R2) and thus critical for the maintenance of dNTP pools for mtDNA synthesis in post-mitotic cells [84], increased 3-4-fold after both short- and long-term treatment with si-hVDAC1. On the other hand, the expression levels of other DNA replication proteins decreased 2- to over 100-fold following long-term treatment with si-hVDAC1 (Figure 11C).

The expression of the La ribonucleoprotein domain family member 7 (LARP7), regulating telomerase activity [85], decreased over 120-fold following long-term tumor hVDAC1 depletion (Figure 8C, Appendix A). Similarly, the expression of the replication factor C subunit 4 (RFC4) regulating telomere maintenance, nuclear DNA replication, mismatch repair, and nucleotide excision repair [86], was reduced 5-fold, while RFC3 levels increased 4-fold (although not significantly) following short-term treatment and decreased over 55-fold after long-term treatment with si-hVDAC1 (Figure 11C, Appendix A).

The expression levels of the Flap endonuclease 1 (FEN1), were unchanged or reduced 2-fold in the short-term si-hVDAC1-TTs and further reduced 3–4-fold in the long-term si-hVDAC1-TTs (Figure 11C). FEN1 regulates DNA replication, repair, recombination and transcription of topoisomerase I (TOP1) [87] and is involved in the regulation of transcription, replication and genome stability [88], and of single-stranded DNA-binding protein (SSBP1), a house-keeping mitochondrial protein controlling mitochondrial biogenesis, mtDNA stability and maintenance [89].

Similarly, the expression levels of the mini-chromosome maintenance proteins MCM3, MCM5, and MCM7, required for replication initiation, modulation of DNA synthesis and control of DNA elongation [90], were reduced 5-6-fold following long-term treatment with si-hVDAC1. 

Thus, except for RRM2B, the expression levels of the group of proteins involved in mitochondrial and nuclear DNA replication decreased following long-term treatment of the GBM tumor with si-hVDAC1. 

## 4. Discussion

Cancer cell survival requires the coordination of transcriptional programs with signaling and metabolic pathways. By down-regulating hVDAC1 expression, the metabolic alterations that happen in tumor cells affect gene expression, and thus biological outcomes in the cancer cell, leading to differentiation. To decipher the mechanisms underlying the metabolic rewiring and consequent tumor reprograming following treatment with si-hVDAC1 [16,18,19,91] we characterized the changes in the expression of selected proteins in tumor sections (Figure 1, Figure 2, Figure 3, Figure 4, Figure 5, Figure 6 and Figure 7). In addition, we performed proteomics analysis to assess global changes. Specifically, we addressed alterations in the expression of proteins associated with signaling, protein synthesis and degradation, DNA structure and regulation, development and differentiation, and cross talk between the cancer cell and its microenvironment and how and why these changes are time-dependent. We demonstrated that reprograming of the transcriptional pattern develops with the length of time the tumor cell has been depleted of hVDAC1, suggesting that the reprograming induced by metabolic limitation involves a chain of events that progressing with time. From the perspective of molecular understanding of cancer, these findings suggest that cancer treatment by any means should consider the time-dependent reprograming elicited by the disease.

Interestingly, we found that reducing the level of VDAC1 expression resulted in an increased expression of pro-apoptotic proteins although no cell death was induced, suggesting a non-apoptotic role for these proteins. In addition, tumors depleted of VDAC1 had a dramatically reduced level of expression of TSPO and the protein was seen to translocate to the nucleus, suggesting a tight link between VDAC1 and TSPO as presented in our recent paper [92].

### 4.1. Metabolic Rewiring upon VDAC1 Depletion

Reprogramed cellular energy metabolism is a hallmark of almost all cancers, irrespective of the origin of the tissue [16,93,94]. The metabolic adaptation of cancer cells provides most cancer cells with precursors for the biosynthesis of nucleic acids, phospholipids, fatty acids, cholesterol, and porphyrins [24,95], all needed for fueling cancer cell division and growth. Reprogramed metabolism and gene deregulation are both hallmarks of cancer [84,96,97,98]. This link between gene expression and metabolism is tightly associated with cellular adaptation of the tumor to nutritional changes during tumor progression. 

The metabolic switch in tumor cells involves metabolic pathways that need to be changed in a coordinated manner to ensure the adaptation of cancer cells to environmental alterations [99]. Here, we showed that depletion of hVDAC1 in GBM cancer xenografts altered the expression of key proteins related not only to metabolism but also to stem cells, differentiation, and to a variety of functions mediated by TSPO, Cyto c, or caspase 3. Furthermore, proteomics analysis revealed proteins related to alterations in the expression levels of transport and trafficking, signaling, development and differentiation, the microenvironment, protein synthesis and degradation, DNA structure and replication and epigenetic regulation (Figure 8, Figure 9, Figure 10 and Figure 11, Appendix A).

The expression levels of metabolism-related proteins involved in glycolysis, the tricarboxylic acid cycle, and OXPHOS were highly modified. Specifically, there was a striking reduction in the levels of HK-I, GAPDH, LDH-A, CS and ATP synthase following both long and short-term treatment with si-hVDAC1 (Figure 1D–M). The expression of ABHD10 and NDUFS8 decreased only after long-term tumor treatment with si-hVDAC1. In contrast, there was an increase in the expression of other proteins such as TIGAR, a negative regulator of glycolysis, aldehyde dehydrogenase 2 (ALDH2) following long-term si-hVDAC1 treatment (Figure 8A). The expression levels of transport mediating proteins, such as the glucose transporter Glut 1 and the zinc transporter ZIP14 (SLC39A14) were greatly reduced following the long-term treatment (Figure 8B and Appendix A).

Glycolysis can be targeted by inhibiting HK activity using 2-deoxyglucose (2-DG), 3-bromopyruvate (3-BP), an alkylating reagent that also targets GAPDH, or lonidamine, a derivate of indazole-3-carboxylic acid [100]. However, potential concerns and challenges have been discussed with respect to the use of glycolytic inhibitors for cancer treatment, as these compounds affect certain normal tissues, including retina and testis, brain, which use glucose as their main energy source [100]. Our results, on the other hand, show that by preventing the expression of VDAC1, the mitochondria gatekeeper protein, one can target global cell metabolism, rather than targeting a single enzyme or pathway.

### 4.2. TSPO Over-Expression in GBM, Nuclear Localization and Regulation by VDAC1 

TSPO has been shown to be overexpressed in a variety of cancers [35,36,37,38] (Figure 5). TSPO has been used for in vivo monitoring of glial cell activation and was proposed as a marker for metastatic growth in the brain [101]. By following [^18^F]DPA-714 uptake, TSPO was shown to be highly expressed in glioma cell lines and tumors [102,103]. In addition, the increased binding of 123I-DPA-713 seen in an intra-cerebrally induced model of brain metastasis suggests an increase in TSPO expression in brain metastasis [104]. All these findings together suggest that TSPO may serve as a biomarker for brain cancer [105].

The role of the overexpressed TSPO in cancer is not clear although it may be related to the cholesterol transport activity of the protein. In this context, many studies have reported that cholesterol accumulation might be a general property of tumor cells [106,107,108,109,110]. TSPO acts as a mediator of cellular proliferation and is involved in metabolism, cell proliferation [111,112], immunomodulation and apoptosis [113], inflammation [34,114], Ca^2+^ signaling [115], and oxidative stress regulation [116]. TSPO over-expression in cancer might thus be required to support these functions. This supposition is supported by the findings that TSPO ligands have also been shown to exhibit anti-proliferative effects in a range of cancer cell types [117]. Moreover, increased TSPO expression could be correlated with a more aggressive cancer cell phenotype [38,118].

Our findings that down-regulation of VDAC1 in glioblastoma also down-regulated TSPO expression [1] (Figure 6) suggests that this should affect the tumor functions supported by TSPO such as metabolism, and cell proliferation [111,112]. The observed decrease in TSPO levels may be a result of the decrease in VDAC1 as an interacting partner or due to some other unknown factors. Interestingly, in endothelial cells, over-expression of TSPO inhibits VDAC1 expression, while silencing of TSPO increases VDAC1 expression [119], suggesting a reciprocal relationship. 

The physical interaction between TSPO and VDAC1 has been demonstrated in several studies [92]. The functional interplay between TSPO and VDAC1 is thought to dictate the efficiency of mitochondrial metabolism and quality control [120] and the function of TSPO in steroidogenesis is also suggested to be linked to interactions with VDAC [121]. TSPO and VDAC1 were also implicated in NLRP3 inflammasome formation [122] where an increase in VDAC1 levels led to activation of the NLRP3 inflammasome, while down-regulation of VDAC levels in THP1 cells resulted in decreased caspase-1 activation and IL-1β secretion following inflammasome activation [123].

Here we found TSPO in the nuclei of patient-derived section and U-87MG-xenograft tumors but not in cultured U-87-MG cells (Figure 6 and Figure 7). Using immunostaining and a fluorescent derivative of the TSPO ligand FGIN-27, revealed TSPO in the nucleus and perinuclear area of the human breast cancer cell line MDA-231 and human breast tumor biopsies [118,124]. Similarly, TSPO was detected in the nuclei of a human glioma cell line and a glioblastoma tumor biopsy, where the presence of TSPO in the nucleus was correlated with tumor aggressiveness [118,124]. Electron microscopy and immunostaining of rat primary astrocytes revealed the presence of peripheral benzodiazepine receptor (PBR)/TSPO in many subcellular locations including the OMM, plasma membrane, ER, nuclear membrane, and the centrioles of dividing cells [125].

Both the nuclear localization of TSPO and cholesterol transport have been shown to increase in glial, breast and prostate metastases [38]. In breast cancer, TSPO expression, nuclear localization, and TSPO-mediated cholesterol transport into the nucleus have been associated with cancer cell proliferation and expression of an aggressive phenotype, with a correlation between TSPO expression and advanced stages of this malignancy [126].

It should be noted that studies using TSPO knockout mice were unable to confirm the essential role of TSPO in the transport of cholesterol from the outer to the inner mitochondrial membrane [127].

### 4.3. Tumor Cell Differentiation Precedes Metabolic Reprograming

We demonstrated that hVDAC1 depletion in tumors altered the expression levels of several CSC-, development-, and differentiation-related proteins in GBM, lung and triple negative breast cancer tumors [16,19,91]. CSCs, a sub-population of cells within the tumor, are considered to be tumor-initiating cells [128]. It was proposed that glioma cells are created from NSCs found at a different commitment state [129]. Therefore, glioma CSCs and NSCs share markers such as CD133, CD44, CD15. In this respect, not only were CD44 and CD133 expression levels reduced in tumors treated for short and long periods with si-hVDAC1 (Figure 2) but so were those of other CSCs markers, such as Sox-2, Oct3/4, Nestin, and KLF4, all shown to be highly reduced upon differentiation [130]. These results suggest the “disappearance” of CSCs, possibly via differentiation into mature neuronal cells. 

Only following long-term treatment with si-hVDAC1 did tumor cells express neuronal markers such as those of astrocytes (i.e., GFAP, TUBB3, and GAD-67) [16] (Figure 3), suggesting that CSCs are differentiated into cells resembling astrocytes or neurons. Thus, we suggest that differentiation into less tumorigenic cells reflects CSCs that were eliminated upon differentiation. It has been proposed that glioma CSCs express differentiation markers similar to those in normal glia and neurons [131], such as of oligodendrocytes (ASCL1, OLIG2 and DLL3), astrocytes (GFAP), neurons (β-tubulin III, SYT1 and SLC12A5), and markers of inflamed astroglial cells (SERPINE1, TGFB1 and RELB) [132]. In addition, glioma CSCs can differentiate into neuron- and glial-like cells [133]. Thus, our findings showing both astrocyte- and neuron-like cells agree with the above reports. 

In contrast to the reprograming of metabolism and the elimination of CSCs that was already observed following the relatively short-term treatment with si-hVDAC1 (Figure 1 and Figure 2), differentiation was obtained only after a long period of the tumor cells being depleted of VDAC1. Thus, we suggest that tumor cell differentiation is a prolonged process that precedes metabolism reprograming and the “disappearance” of cancer stem cells.

Interestingly, it has been shown that miR-29-mediated control of VDAC1 expression is associated with the survival of neuronal cell in the brain [134].

Transforming CSCs/cancer cells into normal-like cells could be a promising approach for treating cancer. Indeed, several studies have focused on the mechanisms by which differentiation of undifferentiated CSCs could be induced. Differentiation of nasopharyngeal carcinoma cells was induced via inhibiting an epigenetic mechanism by which EZH2 (enhancer of zeste 2 polycomb repressive complex 2 subunit) represses IKKα (inhibitor of nuclear factor kappa-B kinase subunit beta) expression and impairs tumor cell differentiation [135]. Treatment of ductal carcinoma in situ relied on efatutazone (a PPARγ agonist), leading cells to be less invasive and show elevated differentiation [136].

VDAC1 deletion was found to up-regulate the expression of the pro-apoptotic proteins, caspase 3, and Cyto c, although there was no induction of apoptosis. These proteins have previously been reported to play a non-apoptotic role associated with cell differentiation in non-cancer cells [137,138], but such functions have not been well established in cancer cells. We suggest that these pro-apoptotic proteins mediate a non-apoptotic role as promoting cell differentiation (25).

AIF expression levels were dramatically decreased in si-hVDAC1-TTs, suggesting an over-expression in untreated tumors. Considering the pro-apoptotic function of AIF several possible explanations as to how cancer cells tolerate the increased AIF expression have been proposed [32]. Here we suggest that AIF over-expression may offer an advantage to cancer cells via its additional non-apoptotic role as a tumor suppressor [139]. This is associated with AIF functions related to oxidative phosphorylation, reactive oxygen species (ROS) detoxification, and maintaining normal mitochondrial redox balance, mitochondrial morphology, and cell cycle regulation [33,140]. Thus, the dramatic decrease in AIF expression in si-hVDAC1-TTs, eliminates these AIF functions, and thereby the ability of tumors to exploit the oxidative and metabolic capabilities of AIF in favor of tumorigenesis.

Here, we showed that tumor depletion of VDAC1 affects the reprograming of cell metabolism, induced differentiation, and led to changes in the expression of transcription factors controlling signaling pathways connected to cancer hallmarks, thereby interfering with the establishment of oncogenic properties, including the tumor microenvironment.

### 4.4. si-hVDAC1 Tumor Treatment Leads to Microenvironment Rearrangement

Cancer development depends on the involvement of normal cells in the tumorigenic process, such as fibroblasts, endothelial cells, and cells of the immune system, which support a “smoldering” of inflammation. Interestingly, although the si-RNA used here was specific to human VDAC1, it affected the expression of proteins of mouse origin associated with stromal activity (Figure 10B). 

Within a tumor, crosstalk between cancer cells and the tumor microenvironment regulates tumorigenesis, as re-modeling of the stroma is essential for cancer development and migration [141]. The tumor stroma is composed of elastin, collagens, fibronectin, laminins, and cellular proteases, such as cathepsins and matrix metalloproteinases [142]. 

The expression of mouse proteins associated with ECM organization and modification, such as EMB, MMP8, and MMP9, all zinc-dependent enzymes [143], was reduced (6-18 fold), while that of myosin-10, which functions in cytoskeleton reorganization and lamellipodial extension, was highly increased (~80-fold), mainly following long-term treatment with si-hVDAC1 (Figure 10B, Appendix A). In addition, the amount of human protein SUSD2, a Notch3 regulating protein that acts in cell adhesion and migration [53], increased over 150-fold, while levels of the endothelial intercellular cell adhesion molecule ICAM1 [54] decreased after long-term si-hVDAC1 treatment (Figure 10A, Appendix A).

The results show that tumor treatment with si-RNA against human VDAC1 resulted in tumor microenvironment reprograming, leading to reduced stromal activity essential for tumorigenesis and inflammation. In other words, VDAC1 depletion in tumor cells of human origin resulted in a global change in cancer cell metabolism, with these metabolic changes affecting the expression of genes associated with the microenvironment of mouse and human cells in the tumor. Thus, metabolism reprograming in the implanted human cancer cells affected gene transcription in mouse host cells.

### 4.5. Rewiring Metabolism Alters Signaling Pathways, as Mediated via Regulation of Protein Synthesis and Degradation, Epigenetics, and Gene Transcription

Cancer cell reprograming, including directing cells towards differentiation, involves molecular signaling pathways and requires transcriptional activity. Our proteomics analysis (Figure 4, Figure 5, Figure 6, Figure 7 and Figure 8, Appendix A) revealed global changes in proteins involved in signaling, protein synthesis and degradation, DNA replication, structure and epigenetics, all of which were highly altered after a long period of si-hVDAC1 treatment. Specifically, we demonstrated that in VDAC1-depleted tumors, the expression of several development- and differentiation-related proteins was highly modified. The expression of ROCK1, a serine protein kinase expressed mostly in immune cells and regulated by the small GTP-binding protein RhoA, affecting cell proliferation, adhesion, inflammation, and oxidative stress, was greatly decreased (~15-fold; Figure 10A).

The expression of PGAM5 that dephosphorylates and activates ASK1 MAP3K [144] and prolongs survival of tumor cells in hepatocellular carcinoma (HCC) [145], and PLXNB2, an activator of the GTPase Rho [56], promoting ovarian cancer cell proliferation and invasion [57] and prostate tumor growth [58], as well as ICAM1, which activates Rho [54] and which increased metastasis in pancreatic ductal adenocarcinoma [55], were reduced (5- and 16-fold). On the other hand, the expression levels of SUSD2 (Figure 10A), a Notch 3-regulating protein that also suppresses metastasis in high-grade serous ovarian carcinoma [146], and of STAT6, playing a major role in regulation of IL-4 synthesis, reducing colon inflammation during tumorigenesis but facilitating tumor formation [147], increased (150- and 12-fold, respectively). Similarly, the expression levels of LARP7, regulating telomerase activity, and RFC3 were reduced roughly 120- and 50-fold, respectively (Figure 11C).

Changes in the expression of various proteins in networks associated with cancer development and survival are mediated by epigenetic considerations, with the interplay between metabolism and epigenetics being well established [148,149,150,151,152,153,154,155,156,157,158]. Epigenetics and gene transcription are highly influenced by products of metabolic pathways, with emerging evidence of crosstalk between processes occurring in the mitochondria and those taking place in the nucleus [159]. Thus, it is not surprising that the reprograming of cancer cell metabolism by hVDAC1 depletion led to global changes in the expression profiles of proteins that are highly enhanced following long-term treatment with si-hVDAC1.

To conclude, down-regulation of hVDAC1 expression in a glioblastoma xenograft resulted in global changes in tumor hallmarks, with reprograming of cell metabolism altering the expression of proteins associated with signaling pathways, protein synthesis and degradation, and CSCs, leading to differentiation. In addition, no apoptotic cell death was induced by VDAC1 depletion, yet the expression levels of proteins associated with apoptosis were altered, suggesting serving non-apoptotic function. Interestingly, the expression of TSPO, a VDAC1 interacting protein with multiple activities was highly decreased. These changes involve a reprograming of the transcriptional pattern that develops with time and which was highly pronounced after a long period of the cells being depleted of VDAC1. This suggests that the reprograming induced by metabolism limitation involves a chain of events that occur in a sequential manner and points to the interplay between metabolism and oncogenic signaling networks.

## Figures and Tables

**Figure 1 cells-08-01330-f001:**
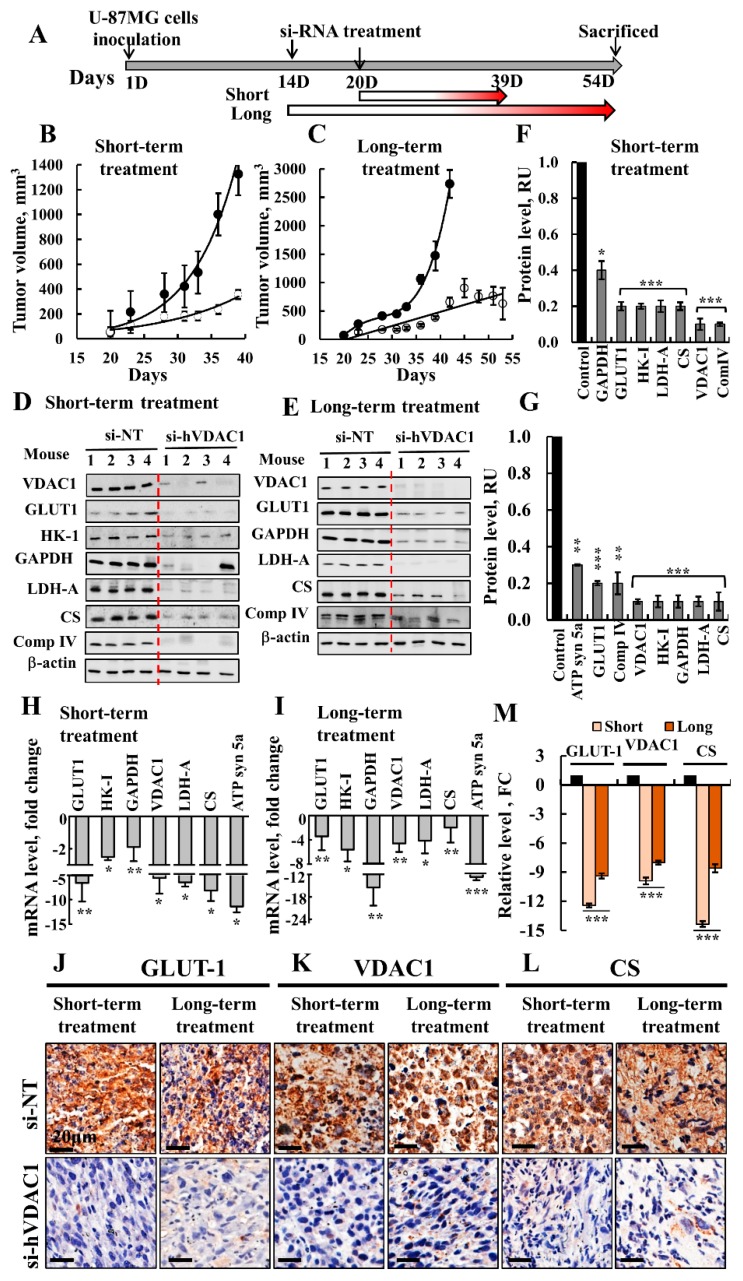
Inhibition of tumor growth and metabolism reprograming in a mouse GBM xenograft model following short- and long-term treatment with si-hVDAC1. (**A**) Schematic representation of the GBM xenograft in a mouse and si-hVDAC1 treatment applied for short and long periods. (B, C) U-87MG cells were s.c. inoculated into nude mice. When tumor volume reached 50–100 mm^3^, the mice were divided into matched 2 groups, and the tumors were treated every 3 days with si-NT (●, 5 mice) or si-hVDAC1 (○, 5 mice) to a final concentration of 50–60 nM for 19 days (**B**) or 40 days (**C**). The calculated average tumor volumes are presented as means ± SEM; *p*: * ≤ 0.05; ** ≤ 0.01; *** ≤ 0.001. Immunoblots (**D**,**E**) and the quantitative analysis (**F**,**G**) of metabolism-related proteins from si-NT-TTs and si-hVDAC1-TTs, treated with si-hVDAC1 for short (**D**,**F**) or long periods (**E**,**G**). (**H**,**I**) q-RT-PCR analysis of the indicated genes presented as fold change in si-NT-TTs and si-hVDAC1-TTs obtained from U-87MG tumors treated for short (**H**) or long periods (**I**). IHC analysis of tumor sections obtained from tumors treated with hVDAC1 or si-NT for short or long periods and stained for Glut1 (**J**), VDAC1 (**K**) and CS (**L**) using specific antibodies with quantitative analysis (**M**). Quantification of the image staining intensity was generated using a panoramic microscope and HistoQuant software (Quant Center 2.0 software, 3DHISTECH Ltd). All data are expressed as Mean ± SEM, (n = 3 tumors) (* *p* < 0.05; ** *p* < 0.01, *** *p* < 0.001).

**Figure 2 cells-08-01330-f002:**
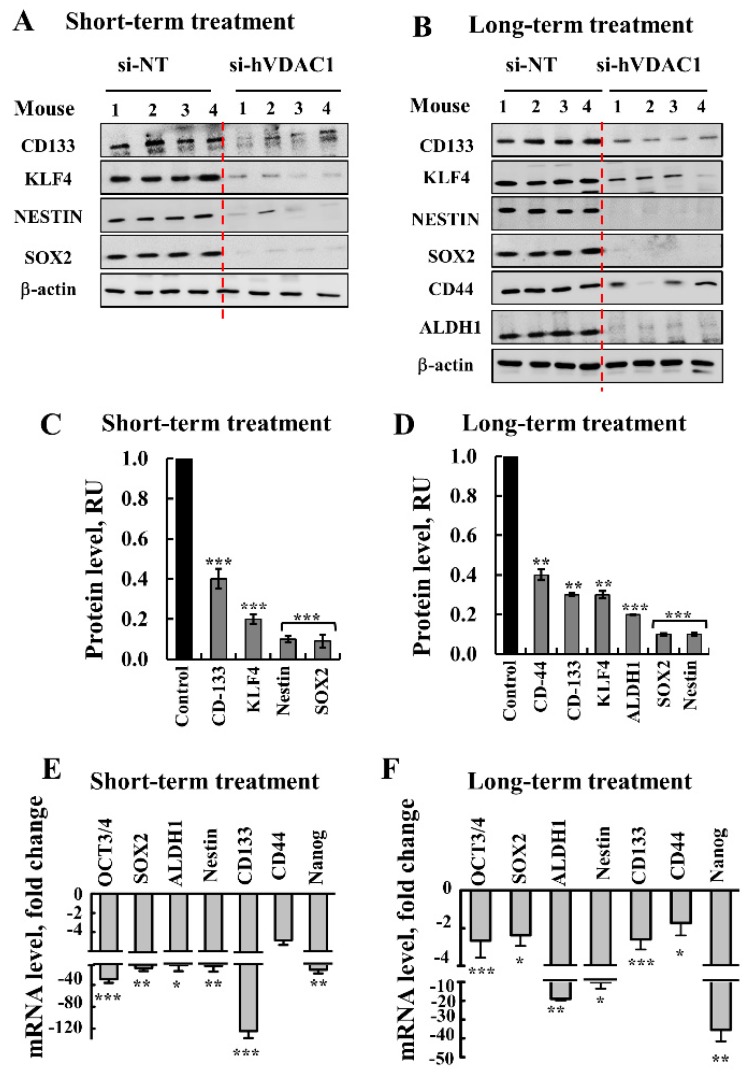
si-hVDAC1 treatment markedly reduces cancer stem cell marker expression in U-87MG-derived tumors. (**A**–**D**) Immunoblot (**A**,**B**) and quantitation (**C**,**D**) of protein extracts obtained from U-87MG-derived xenografts treated with si-NT or si-hVDAC1 short (**A**,**C**) or long term (**B**,**D**), using antibodies specific to the indicated GBM stem cell markers. β-actin is used as an internal loading control. (**E**,**F**) mRNA levels of the indicated genes in si-hVDAC1-TTs, relative to those in si-NT-TTs derived from U-87MG tumors subjected to short-term (**E**) or long-term (**F**) treatment. All data are expressed as Mean ± SEM, (n = 5 tumors) (* *p* < 0.05; ** *p* < 0.01, *** *p* < 0.001).

**Figure 3 cells-08-01330-f003:**
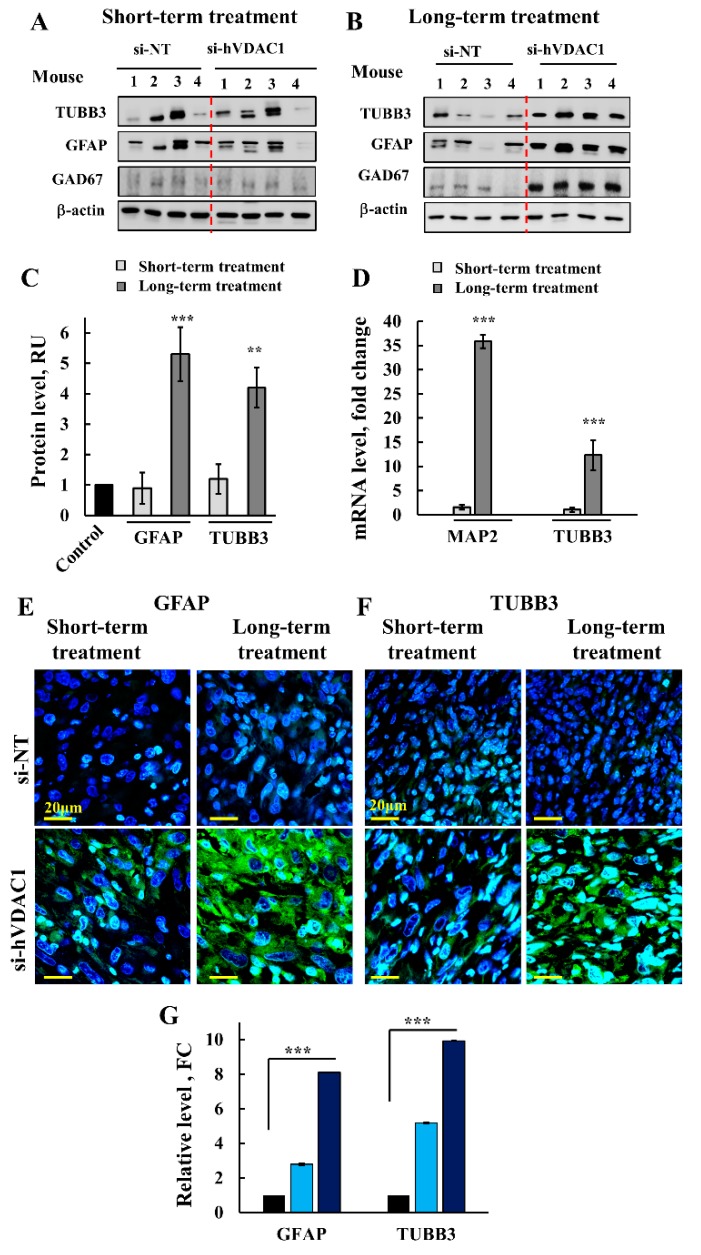
si-hVDAC1 treatment of U-87MG-derived tumors induces the expression of proteins associated with differentiation only after long-term treatment with si-hVDAC1. U-87MG cell-derived tumors treated with hVDAC1 or si-NT for short or long periods were subjected to immunoblotting (**A**,**B**) using antibodies against proteins associated with differentiation (glia fibrillary acidic protein (GFAP), TUBB3, and GAD-67), and quantitative analysis of the blots (**C**,**D**) mRNA levels of the indicated genes were evaluated by q-RT-PCR and appropriate primers and the results are presented relative to those in si-NT-TTs. (**E**,**F**) Tumor sections from hVDAC1-TTs or si-NT-TTs following short- or long-term treatment were subjected to immunofluorescence staining for GFAP (**E**) and TUBB3 (**F**) followed by staining quantification (**G**) using Image J. The results are presented as fold change (FC) relative to the values in si-NT-TTs. All data are expressed as mean ± SEM, (n = 3 tumors); ** *p* < 0.01, *** *p* < 0.001).

**Figure 4 cells-08-01330-f004:**
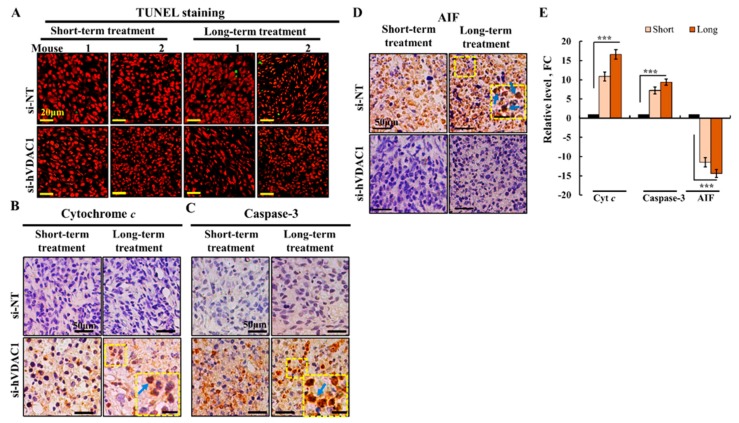
si-hVDAC1-TTs showed no apoptotic cell death, despite changes in the expression levels of pro-apoptotic proteins. (**A**) Paraffin-embedded sections from hVDAC1-TTs or si-NT-TTs following short- or long-term treatment were subjected to TUNEL staining, as described previously [1]. (**B**) Representative sections from si-NT-TTs and si-hVDAC1-TTs IHC-stained for Cyto c (**B**), caspase 3 (**C**), and apoptosis-inducing factor (AIF) (**D**), using specific antibodies, and quantitation of the staining intensity (**E**) using a panoramic microscope and HistoQuant software (Quant Center 2.0 software,1.15.1 RTM 3DHISTECH Ltd) are shown as fold change (FC) relative to the levels in si-NT-TTs Inset shows nuclear localization, marked by arrows, bar = 20 μm. Data are expressed as mean ± SEM, (n = 3 tumors), (*** *p* < 0.001).

**Figure 5 cells-08-01330-f005:**
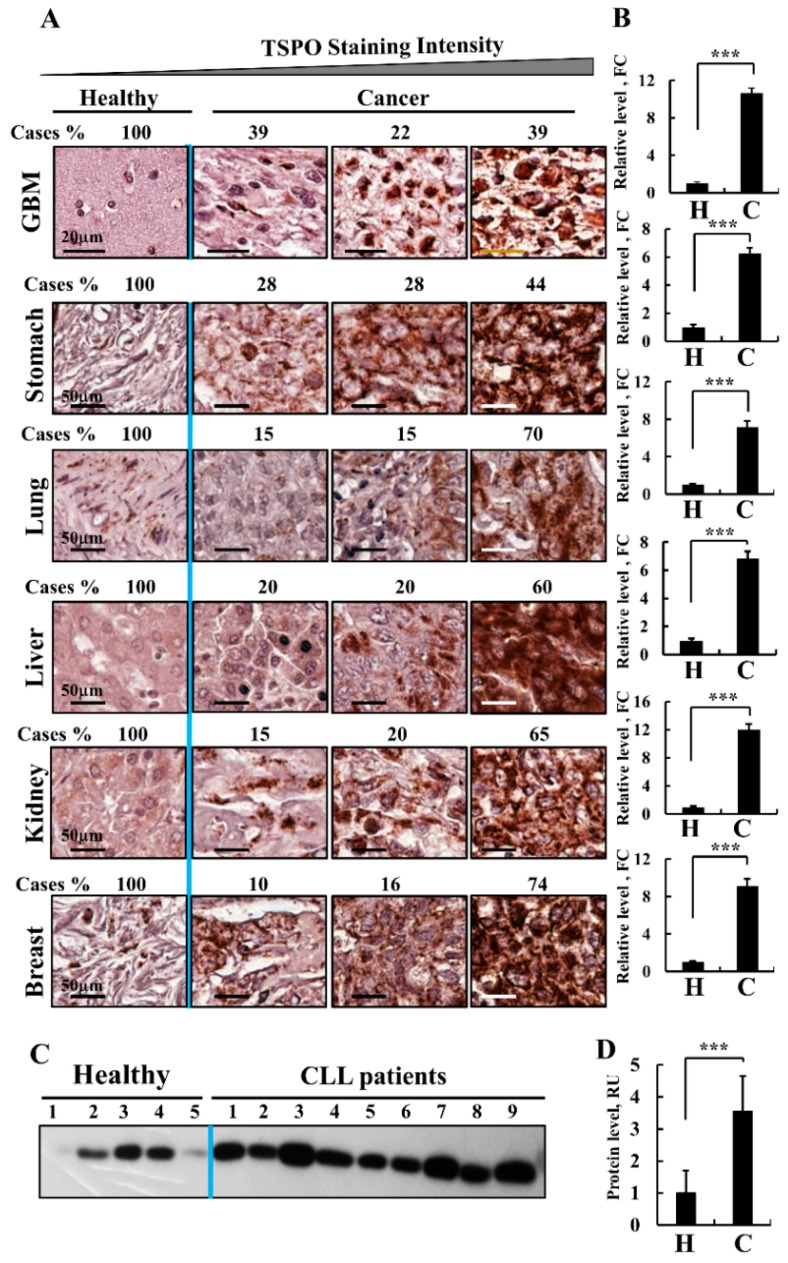
TSPO over-expression in cancer. (**A**) Immunohistochemical staining of TSPO was performed on tissue microarray slides obtained from Biomax US (MC5003, https://www.biomax.us/MC5003b). Representative sections of the indicated tissues from healthy (5 samples for each tissue type) and cancer-derived samples, GBM (12), stomach, lung (10 adenocarcinoma and 9 squamous), liver (each 20), kidney (15) and breast (15). (**A**) The slides were incubated with anti-TSPO antibodies diluted in 1% BSA in PBS overnight at 4 °C and then with secondary antibodies diluted in 1% BSA in PBS. The slides were subsequently treated with 3′3-diaminobenzidine tetra-hydrochloride (DAB) and counter-stained with hematoxylin. Negative controls were incubated without primary antibody. Sections of tissue were observed under an Olympus microscope and images were taken at 200× magnification with the same light intensity and exposure time. TSPO expression levels in healthy donors and cancer patient sections were sub-grouped according to staining intensity into 3 groups with the percentages of sections stained at the intensity indicated by the scale indicated above. (**B**) Quantitation of the staining intensity was performed using a panoramic microscope and HistoQuant software (Quant Center 2.0 software, 3DHISTECH Ltd). (**C**) Representative immunoblot probed with anti-TSPO antibodies showing TSPO over-expression in PBMCs derived from CLL patients (n = 9), as compared to PBMCs from healthy donors (n = 5). For CLL samples details see [2], (*** *p* < 0.001).

**Figure 6 cells-08-01330-f006:**
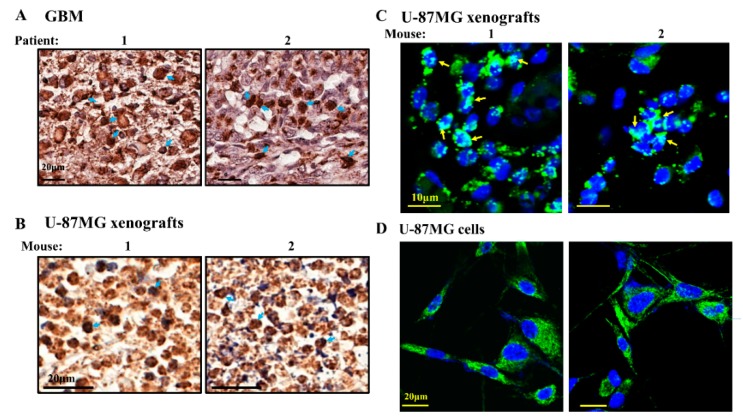
TSPO expression levels decreased in si-hVDAC1 treated U-87MG-derived tumors. Representative sections from si-NT-TTs and si-hVDAC1-TTs, treated for short or long periods were immunohistochemically stained for TSPO (**A**) and the staining intensity was quantified using a panoramic microscope and analyzed by HistoQuant software (Quant Center 2.0 software, 3DHISTECH Ltd) (**B**) are shown as the intensity in relative units (RU). Data are expressed as mean ± SEM, (n = 3 tumors). Confocal fluorescence microscopic images of immuno-stained sections from si-NT-TTs and si-hVDAC1-TTs, treated for short (**C**) or long (**D**) periods stained with anti-TSPO or anti-VDAC1 antibodies. Nuclei were stained by DAPI. Enlargement of the corresponding squad area is presented with arrows indicating the nuclear localization of TSPO.

**Figure 7 cells-08-01330-f007:**
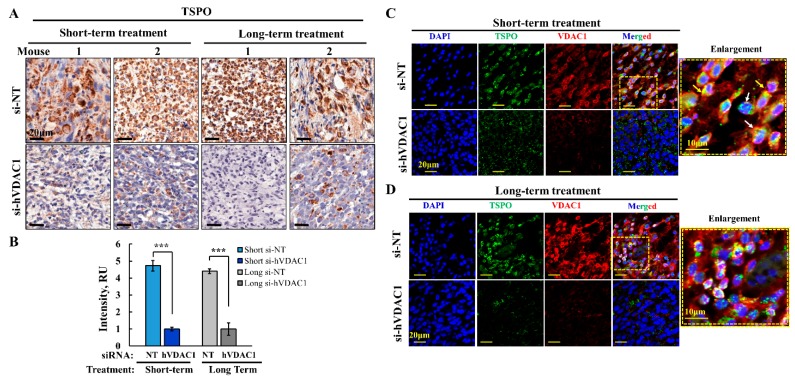
TSPO nuclear localization. (**A**) Representative sections derived from GBM patients from tissue microarray slides (Biomax US (MC5003)), immunohistochemically stained for TSPO with arrows indicating the nuclear localization of TSPO. (**B**,**C**) Representative microscopic images of sections from tumors of 2 mice treated for a long period with si-hVDAC1, IHC (**B**) or IF (**C**) stained with anti-TSPO. Arrows indicate the nuclear localization of TSPO. (**D**) U-87GM cells were grown on coverslips and assayed after 48 h for TSPO using specific antibodies. Nuclei were stained with DAPI, (*** *p* < 0.001).

**Figure 8 cells-08-01330-f008:**
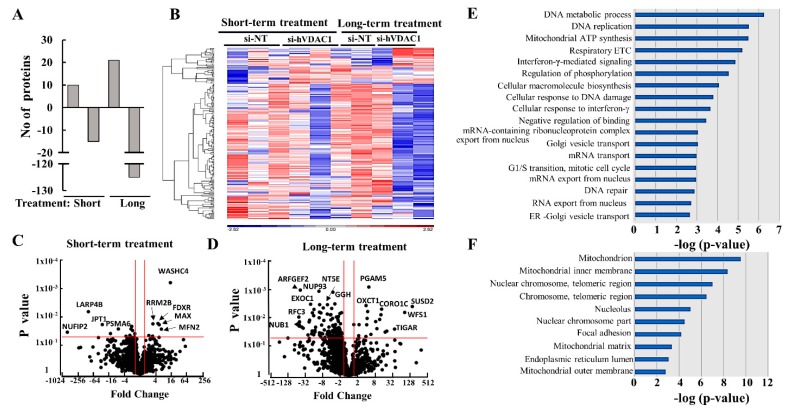
Differentially expressed proteins in si-hVDAC1-TTs and si-NT-TTs subjected to short- and long-term treatment. (**A**) Human proteins differentially expressed between tumors treated with si-NT or si-hVDAC1 for short or long periods, as analyzed by LC-HR-MS/MS. Out of 1,845 human proteins presenting at least 1 unique peptide, those with a *p*-value of 0.05 and linear fold change of +/− 1.5 are presented. (**B**) Volcano plots showing *p*-values as a function of fold change in si-hVDAC1-TTs treated short (**B**) or long term (**C**), relative to si-NT-TTs. Cutoff lines shown are as in (**A**), namely a *p*-value of 0.05 and a linear fold change of +/−1.5. (**D**) Hierarchical clustering of the 167 proteins found to be differentially expressed after either short- or long-term treatment. The color scale of the standardized expression values is shown. (**E**,**F**) Significantly enriched functional groups based on GO biological process (**E**) and GO cellular component (**F**) in the long-term si-hVDAC1-TTs. Shown are GO terms with a Fisher’s exact test FDR-adjusted *p*-value < 0.05 and more than 4 differentially expressed proteins. The X-axis shows the –log10 of the *p*-value.

**Figure 9 cells-08-01330-f009:**
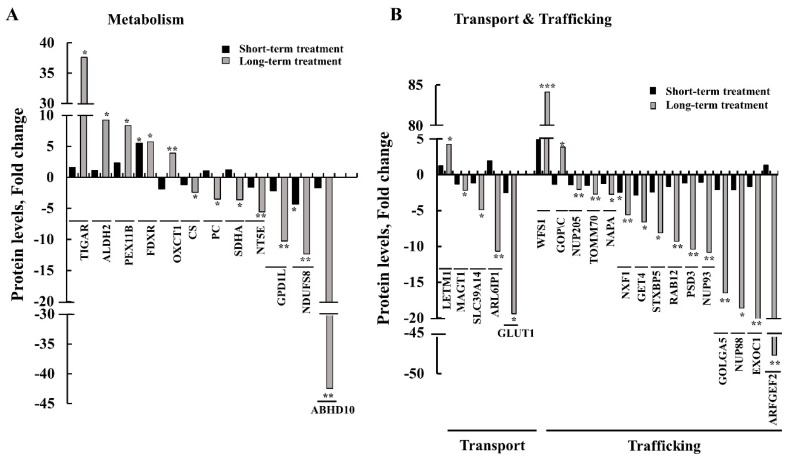
Differentially expressed metabolism-, transport-, and trafficking-related proteins in tumors treated for short or long periods with si-hVDAC1. Quantitative analysis of the LC-HR MS/MS data. Differentially expressed metabolism-related proteins (**A**) and transport- and trafficking-related proteins (**B**) in tumors treated with si-hVDAC1 for short and long periods are presented in terms of fold change of expression in si-VDAC1-TTs, relative to si-NT-TTs. * *p* ≤ 0.05, ** *p* ≤ 0.01. (■) and (

) indicate short-term and long-term treatment with si-hVDAC1, respectively.

**Figure 10 cells-08-01330-f010:**
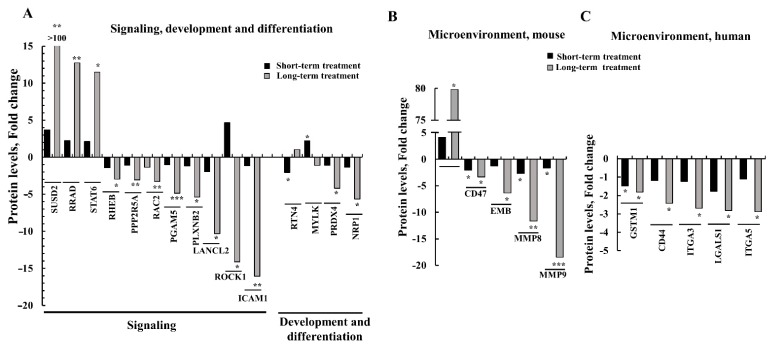
Tumor treatment for long periods with si-hVDAC1 alters the expression of signaling-, development-, differentiation-, and microenvironment-related proteins, relative to their expression following short-term treatment. Quantitative analysis of the LC-HR MS/MS data. (**A**) Differentially expressed proteins associated with signaling, development, and differentiation between tumors treated with si-hVDAC1 for short and long periods are presented as fold change of the expression in si-VDAC1-TTs, relative to si-NT-TTs. Differentially expressed proteins of mouse (**B**) and human (**C**) origin associated with the microenvironment in tumors treated with si-hVDAC1 for short and long periods, presented as fold change of the expression in si-hVDAC1-TTs, relative to si-NT-TTs. (■) and (

) indicate short-term and long-term treatment with si-hVDAC1, respectively. * *p* ≤ 0.05, ** *p* ≤ 0.01, *** *p* ≤ 0.001.

**Figure 11 cells-08-01330-f011:**
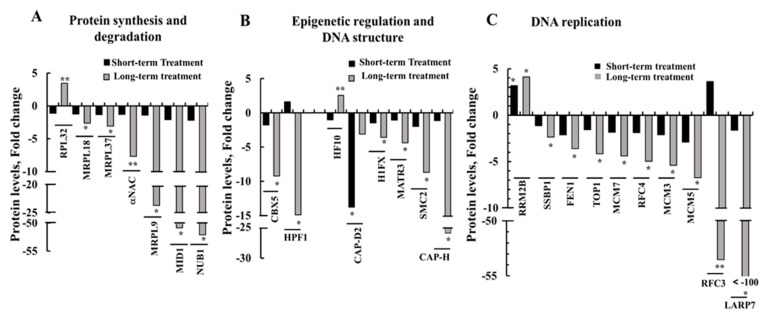
Differentially expressed protein synthesis and degradation-, epigenetic regulation, DNA structure-, and DNA replication-related proteins following short- and long-term treatments with si-VDAC1 or si-NT. Quantitative analysis of the LC-HR MS/MS data. Differentially expressed proteins of mouse (**A**) and human (**B**) origin associated with the microenvironment in tumors treated with si-hVDAC1 for short and long periods, are presented as fold change of the expression in si-VDAC1-TTs, relative to si-NT-TTs. (■) and (

) indicate short-term and long-term treatment with si-hVDAC1, respectively. * *p* ≤ 0.05, ** *p* ≤ 0.01.

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
