# Peer review of "Rewiring of Cancer Cell Metabolism by Mitochondrial VDAC1 Depletion Results in Time-Dependent Tumor Reprogramming: Glioblastoma as a Proof of Concept"

_cells, 2019, doi:10.3390/cells8111330_

Round 1

Reviewer 1 Report

In this article Shoshan-Barmatz et al. investigated the role of VDAC1 depletion in reprograming metabolic and oncogenetic signaling, with an emphasis on how these processes are affected by short term and long term depletion of VDAC1 in glioblastoma xenographs.  

In general this article was well constructed, and extensively documents the effects of VDAC depletion on tumor metabolism, differentiation, apoptotic signaling, and biomarkers though the use of a combination of optical, biochemical and bioinformatic techniques. Additionally this article provides insight on the sequencing of VDAC1 depletion dependent metabolic reprograming by providing proteomic profiles of short and long term treatment groups.

This article is recommended for publication in Cells, once the fallowing items have been addressed

General Items:

1.Western blots.  Blots should be able to be viewed in full resolution.  A simple solution would be to make importants full resolution blots available in the supplement.

Steroidogenesis. While Shoshan-Barmatz et al. mention the role that both this complex and others relating to TSPO play in steroidogenesis this section needs to be further developed in the discussion in relation to ontogenetic outcomes.

Grammatical and Typographical:

Abstract: the sentence “The mitochondrial gatekeeper, the voltage-dependent anion channel 1 (VDAC1) is mediating transport of metabolites and ions in and out of the mitochondria, and is involved in mitochondria-mediated apoptosis.”  Is confusing and should be reorganized as fallows;

“The mitochondrial gate keeper, voltage-dependent anion channel 1 (VDAC1), mediates transport of metabolites and ions in and out of the mitochondria , and is involved in mitochondria- mediated apoptosis.”

Author Response

General Items:

1.Western blots.  Blots should be able to be viewed in full resolution.  A simple solution would be to make importants full resolution blots available in the supplement.

We believe it is important to have the blots in the main text along with  IHC and q-PCR. To address this reviewers' comment, we have increased the size of the blots in Fig.1D, Fig. 2A,B, Fig. 3 A,B.

Steroidogenesis. While Shoshan-Barmatz et al. mention the role that both this complex and others relating to TSPO play in steroidogenesis this section needs to be further developed in the discussion in relation to ontogenetic outcomes.

To address this suggestion, we have searched for published studies related to TSPO steroidogenesis and the relation to ontogenetic outcomes, and we found a single paper (Mercer, K. A., Weizman, R., & Gavish, M. (1992). Ontogenesis of Peripheral Benzodiazepine Receptors: Demonstration of Selective Up-Regulation in Rat Testis as a Function of Maturation. Journal of Receptor Research, 12(4), 413–425). Considering this and that the topic is not in my expertise, I do not feel comfortable to develop such a discussion.

Grammatical and Typographical:

Abstract: the sentence “The mitochondrial gatekeeper, the voltage-dependent anion channel 1 (VDAC1) is mediating transport of metabolites and ions in and out of the mitochondria, and is involved in mitochondria-mediated apoptosis.”  Is confusing and should be reorganized as fallows;

“The mitochondrial gate keeper, voltage-dependent anion channel 1 (VDAC1), mediates transport of metabolites and ions in and out of the mitochondria, and is involved in mitochondria-mediated apoptosis.”

We have modified the sentence in the abstract to the suggested sentence

Reviewer 2 Report

It is an extensive and well executed study which has plenty of material.

Text is well structured and written in a clear and sound language. 

Conclusions follow the results and there are no major issues with this submission as far as I can see.

I would suggest that the authors move information about the siRNA into the main text from the supplement because it would be more visible there, while it is an important aspect of the study. 

There are also a couple of repetitions which can be avoided.

First, the last paragraph in the introduction which in essence is a summary is not needed, it is pretty much repeated in the discussion. I would suggest to trim it down.

The abstract seems to have an error in this sentence:

"The translocator protein Depletion of hVDAC1 greatly reduced the"

perhaps it was meant to say that knock down of hVDAC1 also had an effect on the translocator but it does not read well.

On page 2 these sentences are somewhat repetitive:

cell, allows transport of metabolites and ions in and out of the mitochondria. VDAC1 transfers metabolites, nucleotides, and ions, including Ca2+, fatty acids, and cholesterol across the OMM [16-18].

It would be interesting to know how this channel can handle all these highly diverse solutes. 

I would also suggest that the authors acknowledge that the cell line they use is only a model of GBM and that it would be important to confirm findings in the freshly isolated GBM cell lines from patients.

Author Response

Comments and Suggestions for Authors

It is an extensive and well executed study which has plenty of material.

Text is well structured and written in a clear and sound language. 

Conclusions follow the results and there are no major issues with this submission as far as I can see.

I would suggest that the authors move information about the siRNA into the main text from the supplement because it would be more visible there, while it is an important aspect of the study. 

 As suggested, we have moved the information about the siRNA used in this study to the Material and Methods section in the main text.

There are also a couple of repetitions which can be avoided.

First, the last paragraph in the introduction which in essence is a summary is not needed, it is pretty much repeated in the discussion. I would suggest to trim it down.

 We have shorten this section significantly to 2 sentences

The abstract seems to have an error in this sentence:

"The translocator protein Depletion of hVDAC1 greatly reduced the"

perhaps it was meant to say that knock down of hVDAC1 also had an effect on the translocator but it does not read well.

Thank you for noting that the first 3 words are present there by mistake, this is now corrected

On page 2 these sentences are somewhat repetitive:

cell, allows transport of metabolites and ions in and out of the mitochondria. VDAC1 transfers metabolites, nucleotides, and ions, including Ca2+, fatty acids, and cholesterol across the OMM [16-18].

We have combined and re-phrased these sentences to prevent repeating the same information.

It would be interesting to know how this channel can handle all these highly diverse solutes.  

We have added VDAC1 pore size (3 nm) and the that VDAC1  transportes molecules as large up to 5 kDa

I would also suggest that the authors acknowledge that the cell line they use is only a model of GBM and that it would be important to confirm findings in the freshly isolated GBM cell lines from patients.

 In our previous published work (Ref 16) with GBM, we used not only U-87MG, but other cell lines (U-251MG, U-118MG, LN-18,C6, and GL-261), as well as  human GBM patient-derived cells as MZ-18 and MZ-327 (PDX), and the glioma-derived stem cell line G7.

We have now indicated this in the Introduction by adding: 

 In our previous study [16], using several GBM cell lines (U-87MG, U-251MG, U-118MG, LN-18,C6, and GL-261) and human GBM patient-derived cells as MZ-18 and MZ-327 and the glioma-derived stem cell line G7, we demonstrated that silencing the expression of VDAC1 using specific si-RNA (si-hVDAC1) dramatically inhibited cell growth.